# A *Francisella tularensis* Chitinase Contributes to Bacterial Persistence and Replication in Two Major U.S. Tick Vectors

**DOI:** 10.3390/pathogens9121037

**Published:** 2020-12-10

**Authors:** Brenden G. Tully, Jason F. Huntley

**Affiliations:** Department of Medical Microbiology and Immunology, University of Toledo College of Medicine and Life Sciences, Toledo, OH 43614, USA; Brenden.Tully@rockets.utoledo.edu

**Keywords:** *Francisella tularensis*, *Dermacentor variabilis*, *Amblyomma americanum*, *Haemaphysalis longicornis*, tick-borne disease

## Abstract

Nearly 100 years after the first report of tick-borne tularemia, questions remain about the tick vector(s) that pose the greatest risk for transmitting *Francisella tularensis* (*Ft*), the causative agent of tularemia. Additionally, few studies have identified genes/proteins required for *Ft* to infect, persist, and replicate in ticks. To answer questions about vector competence and *Ft* transmission by ticks, we infected *Dermacentor variabilis* (*Dv*)*,*
*Amblyomma americanum* (*Aa*), and *Haemaphysalis longicornis* (*Hl;* invasive species from Asia) ticks with *Ft*, finding that although *Aa* ticks initially become infected with 1 order of magnitude higher *Ft*, *Ft* replicated more robustly in *Dv* ticks, and did not persist in *Hl* ticks. In transmission studies, both *Dv* and *Aa* ticks efficiently transmitted *Ft* to naïve mice, causing disease in 57% and 46% of mice, respectively. Of four putative *Ft* chitinases, *FTL1793* is the most conserved among *Francisella* sp. We generated a Δ*FTL1793* mutant and found that Δ*FTL1793* was deficient for infection, persistence, and replication in ticks. Recombinant *FTL1793* exhibited chitinase activity in vitro, suggesting that *FTL1793* may provide an alternative energy source for *Ft* in ticks. Taken together, *Dv* ticks appear to pose a greater risk for harboring and transmitting tularemia and *FTL1793* plays a major role in promoting tick infections by *Ft*.

## 1. Introduction

*Francisella tularensis* (*Ft*) is a Gram-negative bacterium and the causative agent of the zoonotic disease tularemia. This facultative intracellular bacterium, lethal to over 300 species, is able to cause a range of flu-like symptoms in humans [1,2]. Due to high morbidity and mortality rates, ease of aerosolization, and low infectious dose, *Ft* has been classified as a Tier 1 Select Agent by the United States (U.S.) Centers for Disease Control and Prevention.

*Ft* is divided into three subspecies, including *Ft* subsp. *tularensis* (Type A), *Ft* subsp. *holarctica* (Type B), and *Ft* subsp. *mediasiatica*, although only Type A and Type B subsp. are virulent to humans. Type A strains are found exclusively in North America, have an LD_50_ of <10 organisms, cause up to 60% mortality if left untreated, and can be further divided into three subpopulations: Type A1a, A1b and A2, with Type A1b causing the most serious infections [3]. Although less virulent, Type B strains still cause disease in humans and are distributed throughout the Northern Hemisphere [4,5]. Through repeated subculturing of *Ft* subsp. *holarctica*, scientists in the former Soviet Union created a live attenuated strain in the 1930s, which has been named the live vaccine strain (LVS). However, LVS is not a licensed vaccine in the U.S. due to unknown mechanism(s) of attenuation, immunization side effects, and only partial protection against Type A aerosol infection [4,5]. Because of decreased biocontainment requirements, LVS has been used by many research groups to study pathogenesis in mice and other animal models [5]. Although interest in bioterrorism-related research has increased due to this classification, approximately half of U.S. tularemia cases are associated with tick bites [6].

Tick-borne disease cases, including Lyme disease, anaplasmosis, Rocky Mountain spotted fever, Powassan virus, and tularemia, have nearly doubled in the U.S. between 2004 and 2016 [7]. Increasing case numbers may be attributed to the geographic range expansion of tick vectors, presenting increased health risks to humans [8,9]. In 2015 alone, 314 cases of tularemia were reported—the highest recorded since 1964, with over 225 cases being reported per year since 2015 [10]. Ulceroglandular tularemia is the most common presentation in the U.S. and is generally attributed to the bite of an infected tick [11]. In the U.S., there are four tick species most commonly associated with tularemia transmission: *Amblyomma americanum* (*Aa*)*, Dermacentor andersoni, Dermacentor occidentalis*, and *Dermacentor variabilis* (*Dv*) [12,13,14,15]. Although ticks have been speculated to play important roles in the environmental persistence and transmission of *Ft* [16], important questions remain about which tick vector(s) pose(s) the greatest risk for transmitting *Ft* and what *Ft* genes are important for infecting and persisting in ticks.

Between 2001 and 2010, three U.S. states (Arkansas, Missouri, Oklahoma) accounted for over 40% of all tularemia cases [17]. However, tularemia cases are widespread throughout the U.S., as highlighted by high numbers of cases from Colorado, Nebraska, South Dakota, and Wyoming in 2015 [10]. The expansive geographic range of *Dv* ticks, present in nearly every state east of the Rocky Mountains, have implicated *Dv* ticks as the main vector for tularemia in the U.S. [6,18]. Furthermore, *Dv* tick bites have been associated with two outbreaks occurring on Martha’s Vineyard, Massachusetts, both of which involved 15 cases of tularemia, and included one fatality [15]. Laboratory studies have demonstrated that *Dv* ticks can acquire *Ft* from infected mice [19] or through capillary tube feeding [20]. However, those studies also demonstrated major differences in the ability of *Ft* to infect, persist, and replicate in *Dv* ticks. Subsequent studies demonstrated that adult *Dv* ticks were able to transmit *Ft* infection to naïve mice, confirming their importance as vectors for *Ft* [18].

Whereas studies from the 1950s detected *Ft* in field-collected *Aa* ticks [21,22], there have not been any contemporary studies to examine *Ft* prevalence in *Aa* ticks. However, it is known that *Aa* ticks are the most abundant species in the south-central U.S., where higher numbers of tularemia cases occur, suggesting that *Aa* ticks also may be major vectors for tularemia [11,12]. Although performed in two separate studies, data from capillary tube-fed ticks indicated that *Aa* ticks may acquire higher *Ft* bacterial numbers than similarly-fed *Dv* ticks [12,20]. However, direct comparisons between tick species were not performed and the applicability of those studies to naturally infected (i.e., blood meal fed) *Aa* and *Dv* ticks is unknown. In addition, direct comparisons of *Ft* transmission from infected *Aa* and *Dv* ticks to naïve animals have not been performed, leaving major questions about which tick vector poses the greatest risk for *Ft* environmental persistence and/or transmission.

*Haemaphysalis longicornis (Hl)*, a human-biting tick native to Asia, has been detected in at least nine U.S. states, including Arkansas, a state which regularly reports the highest number of tularemia cases in the U.S. [11,23]. Studies in Asia have shown that *Hl* experimentally acquires *Anaplasma*, *Babesia*, *Borrelia*, *Ehrlichia*, and *Rickettsia* infections, all of which circulate zoonotically in the U.S. [7,24]. However, the ability of *Hl* to acquire and transmit *Ft* has not been studied.

To provide new information about which tick vector(s) may pose the greatest risk for harboring and transmitting tularemia, this study directly compared the infection, persistence, and replication of *Ft* in *Dv*, *Aa,* and *Hl* ticks using a mouse–tick–*Ft* infection model. *Dv* and *Aa* ticks were selected for these studies because they are most commonly associated with U.S. tularemia cases and both of their geographic ranges overlap with the foci of tularemia infections in the south-central U.S. [6]. We also examined *Ft* infections of *Hl* ticks, given their invasion into the U.S. and questions about their ability to vector various diseases. Despite *Aa* ticks being initially colonized by significantly higher bacterial numbers, *Ft* persisted in and replicated more robustly in *Dv* ticks during our studies (up to 14 weeks). Interestingly, *Ft* was unable to persist in *Hl* ticks, indicating that it is unlikely to serve as a major vector for tularemia. In transmission studies, both *Dv* and *Aa* adult ticks efficiently transmitted *Ft* infections to naïve mice, with all mice succumbing to disease within seven days after tick infestation (two days after ticks completed their blood meal). Although not significant, *Dv*-infested mice had higher bacterial burdens in their blood, skin, and lungs, compared to *Aa*-infested mice. Chitin is a major component of the tick cuticle and, as such, tick-borne pathogens have been speculated to use chitinases as an alternative energy source. Although *Ft* has been reported to encode at least four putative chitinases [25], their roles in tick infections remain unknown. LVS gene locus *FTL1793, chiD,* contains a GH18 chitinase-like domain and is the most conserved putative chitinase among *Francisella* sp. To provide new information about *Ft* genes required for tick infections, we selected *FTL1793* for further study. Recombinant *FTL1793* was expressed and purified and in vitro assays with purified chitin confirmed that *FTL1793* exhibits chitinase activity. An *Ft* gene deletion mutant, Δ*FTL1793,* was generated and used in mouse–tick–*Ft* infections, finding that *FTL1793* was required for full infectivity and persistence of *Ft* in both *Dv* and *Aa* ticks, particularly during the time when nymphal ticks molt to adults (between weeks 4 and 6 after a blood meal). Taken together, these studies confirm that both *Dv* and *Aa* ticks serve as reservoirs and vectors for *Ft*. However, our results indicate that *Dv* ticks may pose a greater risk for tularemia transmission; *Ft* replicates more robustly in *Dv* ticks, and *Dv* ticks transmit higher *Ft* numbers to naïve animals. Finally, our results demonstrated that *FTL1793* chitinase activity is important for *Ft* to survive and persist in ticks. *Ft* chitin degradation in ticks may provide an alternative energy source as this bacterium awaits transmission to a new mammalian host.

## 2. Results

### 2.1. Low Dose Ft Infects, Persists, and Replicates in Dv and Aa Ticks But Is Cleared from Hl Ticks

A previous study described a mouse–tick–*Ft* infection model to reproducibly infect nymphal *Dv* ticks by feeding ticks on *Ft* LVS-infected mice. When comparing infectious doses of 10^5^ to 10^8^ colony forming units (CFU)/mouse, that study found that infectious doses of 10^7^ and 10^8^ CFU/mouse were saturating for *Dv* ticks (no differences in CFU/tick between the two highest doses) but >10^5^ CFU/mouse (2.2 × 10^2^ CFU/tick) was needed for *Ft* LVS to infect, be maintained in, and replicate in nymphal *Dv* ticks [19]. Despite the importance of that study for understanding *Ft* infection and replication in *Dv* ticks, there still are major questions about whether different tick species (e.g., *Dv* and *Aa*) pose distinct risks as *Ft* reservoirs or if invasive tick species (e.g., *Hl*) can serve as new public health threats for transmitting tularemia. As such, the goal of this study was to compare and contrast *Ft* infections in three distinct tick species: *Dv*, *Aa*, and *Hl*. Many variables needed to be considered for this mouse–tick–*Ft* infection model, including the route of mouse infection that results in bacteremia, the time-to-death for bacteremic mice (approx. 36–48 h), and differences in tick blood meal feeding times (approx. 4–6 days). Given these considerations, we established a similar mouse–tick–*Ft* LVS infection model in our laboratory, where nymphal ticks (either *Dv*, *Aa*, or *Hl*) were placed onto mice three days before *Ft* infection (day 5; 12 ticks/mouse; ticks contained in a chamber on each mouse back), mice were i.v.-infected with *Ft* LVS (day 2), ticks continued taking a blood meal for approx. two more days (until day 0), and ticks were harvested when replete (five day total blood meal) or when mice were moribund.

In the first series of studies, we infected mice with 10^5^ CFU/mouse (1.4 × 10^5^ CFU first experiment; 2.3 × 10^5^ CFU second experiment), to directly compare our findings with the previous *Dv*–*Ft* infection study [19] and to assess whether threshold bacterial doses were needed to infect, be maintained in, and replicate in *Aa* and *Hl* ticks. On day 0, comparing groups of mice that had different tick species fed on them, there was no significant difference in *Ft* numbers in blood samples from infected mice (means: 3.1 × 10^5^ CFU/mL for *Dv* mice, 3.9 × 10^5^ CFU/mL for *Aa* mice, and 3.3 × 10^5^ CFU/mL for *Hl* mice; Figure 1), highlighting the reproducibility of our model and demonstrating modest (approx. 0.2 orders of magnitude) *Ft* replication in mouse blood over two days. Subsets of replete ticks from all three tick species were homogenized on day 0, serially diluted, and plated to quantitate bacterial numbers/tick. On day 0, *Aa* ticks were found to contain significantly more (>1 order of magnitude) *Ft* than either *Dv* or *Hl* ticks (means: 5.6 × 10^4^ CFU/*Aa* tick*,* 2.6 × 10^3^ CFU/*Dv* tick, and 2.1 × 10^3^ CFU/*Hl* tick; Figure 1). To account for the possibility that higher *Ft* numbers in individual ticks or tick species (e.g., *Aa* ticks) could be due to differences in tick blood meal volumes, we also analyzed CFU/mg tick weight, finding that *Aa* ticks still harbored significantly higher *Ft* numbers on day 0, compared to *Dv* or *Hl* ticks (Appendix A). As such, 1 order of magnitude higher *Ft* numbers in *Aa* ticks on day 0 appear to be due to inherent differences in *Aa* ticks and not larger acquired blood volumes by this tick species.

Two weeks later (week 2), *Ft* numbers remained significantly higher in *Aa* ticks (mean 2.7 × 10^4^ CFU/*Aa* tick), compared to *Ft* numbers in *Dv* (mean 3.9 × 10^3^ CFU/*Dv* tick) and *Hl* ticks (mean 1.3 × 10^3^ CFU/Hl tick) (Figure 1). As described above, when accounting for tick weight in week 2, *Ft* numbers remained significantly higher in *Aa* ticks (CFU/mg of tick), compared to either *Dv* or *Hl* ticks (Appendix A) Interestingly, between day 0 and week 2, *Ft* replicated 1.5-fold in *Dv* ticks, while decreasing 4.8-fold in *Aa* ticks and 1.9-fold in *Hl* ticks (Figure 1).

At week 4, *Ft* numbers were not significantly different between *Dv* (mean 6.1 × 10^4^ CFU/*Dv* tick) and *Aa* (mean 1.5 × 10^5^ CFU/*Aa* tick) ticks (Figure 1). Between weeks 2 and 4, *Ft* replicated 11-fold in *Dv* ticks and 5.5-fold in *Aa* ticks (Figure 1). In addition, while *Ft* numbers were fairly consistent within each tick species in weeks 0 and 2, a wide range of bacterial numbers were detected in both *Dv* and *Aa* ticks in week 4 (range 0 to 6 × 10^5^ CFU/tick) (Figure 1). Importantly, *Ft* was not detected in *Hl* ticks in week 4 (Figure 1). Between weeks 2 and 4, all *Hl* ticks molted from nymphs to adults, with all *Hl* ticks surviving the molt. Given that *Ft* was not detected in any *Hl* ticks in week 4, these results indicate that *Ft* may not be able to persist longer than four weeks in *Hl* ticks, *Hl* molting induces changes that result in either clearance or loss of *Ft* infection, or the initial (day 0) *Ft* infectious dose of 2.1 × 10^3^ CFU/*Hl* tick is below the threshold for long-term *Ft* persistence in *Hl* ticks.

Between weeks 4 and 6, >99% of *Dv* and *Aa* ticks molted from nymphs to adults, with no adverse effects of *Ft* infection observed for either tick species (compared to non-infected *Dv* and *Aa* ticks; data not shown). Between weeks 4 and 6, average *Ft* numbers slightly declined in both *Dv* and *Aa* ticks (1.1-fold decrease in *Dv* ticks; 1.5-fold decrease in *Aa* ticks; week 6 means: 5.4 × 10^4^ CFU/*Dv* tick and 6.7 × 10^4^ CFU/*Aa* tick), with no significant difference between *Ft* numbers in *Dv* and *Aa* ticks. Although not significant, 46% of *Dv* ticks (6 of 13 *Dv* ticks) were infected with *Ft* in week 6, compared to 64% of *Aa* ticks (9 of 14 *Aa* ticks) that were infected with *Ft* in week 6 (Figure 1). Despite differences in infection rates, *Dv* ticks were more consistently infected with *Ft* in week 6 (range 4 × 10^4^ to 3.5 × 10^5^ CFU)*,* compared to *Aa* ticks (range 1 × 10^2^ to 5.8 × 10^5^ CFU) (Figure 1). Finally, confirming the week 4 results, *Ft* was not detected in any *Hl* ticks in week 6. Compared to a previous mouse–*Dv* tick–*Ft* infection study [19], our mouse–tick–*Ft* infection model is much more efficient, delivering >1 order of magnitude more *Ft*/tick at the 10^5^ CFU/mouse infectious dose. Our results demonstrated that *Ft* infections as low as 2.6 × 10^3^ CFU/tick can persist and replicate in both *Dv* and *Aa* ticks for up to six weeks. Whereas *Ft* replicated 20-fold in *Dv* ticks between day 0 and week 6, the *Ft* replication rate in *Aa* ticks during the same time frame was only 1.1-fold (Figure 1), indicating that *Dv* ticks may be a better vector for *Ft* replication over six weeks. However, when considering percentages of infected ticks over time, higher percentages of *Aa* ticks (64%) harbored *Ft*, compared to *Dv* ticks (46%). Finally, the highest bacterial numbers in *Dv* and *Aa* ticks were observed in week 4 (4.8 × 10^5^ CFU/*Dv* tick; 6.5 × 10^5^ CFU/*Aa* tick; Figure 1), indicating that there may be a limit to *Ft* replication and bacterial numbers in both *Dv* and *Aa* ticks.

### 2.2. High Dose (10^7^ CFU) Ft Infects, Persists, and Replicates in Dv and Aa Ticks But Is Cleared from Hl Ticks

*Ft* numbers in the blood of naturally infected animals, including mice and rabbits, have not been reported, therefore questions remain about biologically relevant bacteremia numbers for experimental tick infections. However, previous studies have noted *Ft* bacteremia of 10^6^ to 10^10^ CFU/mL in the blood of experimentally infected mice [26,27]. A previous study noted that mouse infectious doses >10^7^ CFU/mL blood did not necessarily result in increased *Ft* numbers in *Dv* ticks [19], suggesting that there may be a limit to the number of bacteria that ticks can support. To test whether higher infectious doses impacted the ability of *Ft* to infect, persist, and replicate in *Dv*, *Aa*, and *Hl* ticks, and to assess whether higher infectious doses could overcome *Ft* clearance/loss in *Hl* ticks, we performed a second series of mouse–tick–*Ft* infection studies using a higher infectious dose of 10^7^ CFU/mouse of *Ft* LVS. Timing of tick placement, intravenous infection, tick harvest, and tick processing were identical to the low dose (10^5^ CFU/mouse) experiments described above, with the exception of adding one additional time point (week 14) to assess longer-term *Ft* persistence in all three tick species, similar to an overwintering event. In these high-dose infection experiments, groups of mice were i.v. infected with 1.3 × 10^7^ CFU/mouse (first experiment) and 2.4 × 10^7^ CFU/mouse (second experiment) of *Ft* on day 2. On day 0, replete ticks (five-day total blood meal) were harvested, and blood was collected from all infected mice. No significant differences were calculated for *Ft* numbers in mouse blood on day 0 when comparing groups of mice that were infested by different tick species (means: 2.1 × 10^8^ CFU/mL for *Dv* mice; 2.6 × 10^8^ CFU/mL for *Aa* mice; 3.4 × 10^8^ CFU/mL for *Hl* mice) (Figure 2). These results highlighted the reproducibility of our mouse infections and demonstrated that, at this higher infectious dose (compared to Figure 1), *Ft* replicated approx. 1 order of magnitude in mouse blood over two days (Figure 2). On day 0, *Aa* ticks were found to contain significantly more *Ft* (0.9 to 1.5 orders of magnitude) than either *Dv* or *Hl* ticks (means: 2.2 × 10^7^ CFU/*Aa* tick, 3.5 × 10^6^ CFU/*Dv* tick, 7.4 × 10^5^ CFU/*Hl* tick; Figure 2). Given that *Aa* ticks harbored significantly higher *Ft* numbers on day 0 at both the lower infectious dose (10^5^ CFU; Figure 1) and at this higher infectious dose (10^7^ CFU; Figure 2), these data suggest that regardless of *Ft* numbers in the blood of infected animals (e.g., 10^5^ or 10^8^ CFU/mL), *Aa* ticks are infected by and harbor more *Ft* on day 0 than either *Dv* or *Hl* ticks. As described above, we calculated CFU/mg of tick to account for potential differences in *Ft* numbers due to inherent differences in tick weights (and/or blood volume), finding that *Aa* ticks still contained significantly higher *Ft* numbers on day 0, compared to *Dv* and *Hl* ticks (Appendix A). In contrast to our 10^5^ CFU infection study findings, at this higher infectious dose, *Dv* ticks were infected by 0.5 orders of magnitude higher *Ft* numbers than *Hl* ticks on day 0 (Figure 2).

Two weeks later (week 2), there were no significant differences in *Ft* numbers among the three tick species, with means of 3.9 × 10^6^ CFU/*Dv* tick, 3.7 × 10^6^ CFU/*Aa* tick, and 1.1 × 10^6^ CFU/*Hl* tick (Figure 2). These results are in contrast to week 2 results from the 10^5^ CFU infection studies, where *Aa* ticks contained significantly higher *Ft* numbers than either *Dv* or *Hl* ticks (Figure 1). At the 10^7^ CFU infectious dose, *Ft* replicated 1.1-fold in *Dv* ticks and 1.4-fold in *Hl* ticks between day 0 and week 2, while decreasing 5.7-fold in *Aa* ticks (Figure 2). Changes in *Ft* numbers in *Dv* and *Aa* ticks at this higher infectious dose were similar to the 10^5^ CFU infection studies, where *Ft* increased >1 order of magnitude in *Dv* ticks and decreased approx. 5-fold in *Aa* ticks between day 0 and week 2 (Figure 1). At this time, we are unable to completely explain the differences between week 2 results when comparing the 10^5^ (Figure 1) and 10^7^ CFU (Figure 2) infectious dose studies, but it remains possible that 10^5^ CFU is near the lower threshold of infectious dose needed to infect *Dv* ticks and/or 10^7^ CFU is near the upper threshold of infectious dose that *Aa* ticks can support. Indeed, the highest *Ft* numbers detected in *Aa* ticks throughout the study were observed on day 0 (Figure 1).

In week 4, no significant differences in *Ft* numbers were calculated among the three tick species, with means of 5.1 × 10^6^ CFU/*Dv* tick, 3.7 × 10^6^ CFU/*Aa* tick, and 2.1 × 10^5^ CFU/*Hl* ticks (Figure 2). Between weeks 2 and 4, *Ft* replicated 1.3-fold in *Dv* ticks, while remaining constant in *Aa* ticks (Figure 2). All *Hl* ticks molted from nymphs to adults between weeks 2 and 4, and *Ft* numbers decreased 5.2-fold in *Hl* ticks during this time frame. Compared to the 10^5^ CFU infection studies (Figure 1), *Hl* ticks did not clear *Ft* by week 4 (Figure 2), indicating that a higher infectious dose (7.4 × 10^5^ CFU/*Hl* tick on day 0) can overcome the previously observed restriction/loss of *Ft* in *Hl* ticks (Figure 1). However, *Ft* only was detected in 50% (4 of 8) of *Hl* ticks in week 4 (compared to 94% infection rate for *Dv* ticks and 95% infection rate for *Aa* ticks), indicating that, even at higher infectious doses, *Hl* ticks are not ideal vectors for tularemia.

In week 6, *Ft* numbers decreased in all three tick species, with means of 2.2 × 10^6^ CFU/*Dv* tick (2.3-fold decrease), 1.8 × 10^6^ CFU/*Aa* tick (two-fold decrease), and 5.1 × 10^3^ CFU/*Hl* tick (42-fold decrease; significantly lower than either *Dv* or *Aa* ticks; *p* < 0.05) (Figure 2). In addition, only 25% (two out of eight) of *Hl* ticks were infected with *Ft* in week 6 (compared to 90% infection rate for *Dv* ticks and 95% infection rate for *Aa* ticks), providing further evidence that *Hl* ticks are not ideal vectors for tularemia. Between weeks 4 and 6, >99% of *Dv* and *Aa* ticks molted from nymphs to adults, with no adverse effects noted in *Ft*-infected ticks compared to non-infected ticks (data not shown).

Finally, a subset of ticks was processed in week 14 to assess the ability of *Ft* to persist in all three tick species long-term, similar to what would occur during an overwintering event in the south-central U.S. (approx. 3.5 months). In week 14, *Ft* numbers were more variable than week 6, however *Ft* replicated 2.7-fold in *Dv* ticks (compared to week 6; mean 6.1 × 10^6^ CFU/*Dv* tick; 85% of *Dv* ticks infected with *Ft*), while *Ft* numbers declined 5.2-fold in *Aa* ticks (compared to week 6; mean 3.4 × 10^5^ CFU/*Aa* tick; 95% of *Aa* ticks infected with *Ft*) (Figure 2). Importantly, *Ft* was not detected in any *Hl* ticks in week 14 (Figure 2) and indicated that, in agreement with our 10^5^ CFU infection results where *Ft* did not persist in *Hl* ticks >4 weeks (Figure 1), *Hl* ticks are unlikely to serve as a major vector for tularemia, regardless of the infectious dose introduced into *Hl* ticks or a later time point examined (>4 weeks). In summary, at this higher infectious dose (10^6^ CFU/*Dv* tick; 10^7^ CFU/*Aa* tick), both *Dv* and *Aa* ticks supported the long-term (14 week) persistence of *Ft*. Despite *Aa* ticks being initially infected by 1 order of magnitude more *Ft* than *Dv* ticks (day 0; Figure 2), bacterial numbers decreased by nearly 2 orders of magnitude in *Aa* ticks over 14 weeks, compared to a modest replication (0.3 orders of magnitude) of *Ft* in *Dv* ticks over the same time period (Figure 2). In contrast, *Hl* ticks do not appear to be a major vector for tularemia; they initially were infected by 1 to 2 orders of magnitude less *Ft* than either *Dv* or *Aa* ticks (day 0) and, although *Ft* modestly (1.4-fold) replicated in *Hl* ticks between day 0 and week 2, *Ft* numbers rapidly declined in *Hl* ticks throughout the remainder of the experiment, with no *Hl* ticks containing *Ft* in week 14. Compared to the 10^5^ CFU infection experiments, where *Ft* numbers increased 20-fold in *Dv* ticks and 1.1-fold in *Aa* ticks over six weeks, infections of *Dv* and *Aa* ticks at this higher infectious dose (10^7^ CFU) revealed no *Ft* replication in *Dv* ticks and a reduction in *Ft* of approx. 1 order of magnitude in *Aa* ticks over six weeks (day 0 to week 6), suggesting that there may be a limit to *Ft* numbers that *Dv* and *Aa* ticks can support. Indeed, the highest number of bacteria detected in these 10^7^ CFU infectious dose studies was 9 × 10^7^ CFU/*Dv* tick (week 14) and 7.5 × 10^7^ CFU/*Aa* tick (day 0) (Figure 2). Given that the highest *Ft* numbers were detected in *Aa* ticks at the beginning of the experiment (day 0) and the highest *Ft* numbers were detected in *Dv* ticks at the end of the experiment (week 14; Figure 2), together with replication rate data (20-fold replication in *Dv* ticks at 10^5^ CFU infectious dose; 2.7-fold replication in *Dv* ticks between weeks 6 and 14 at 10^7^ CFU infectious dose), our results indicate that of the three tick species examined here, *Dv* ticks pose the greatest risk for *Ft* persistence and replication.

### 2.3. Ft Is Efficiently Transmitted by Infected Dv and Aa Ticks to Naïve Mice

We next assessed the ability of infected *Dv* and *Aa* ticks to transmit *Ft* to naïve mice. *Hl* ticks were not included in transmission studies because of lower initial infection rates (compared to *Dv* and *Aa* ticks), clearance/loss of *Ft* in *Hl* ticks in low dose (10^5^ CFU) infection studies by week 4 (Figure 1), and clearance/loss of *Ft* in *Hl* ticks in high dose (10^7^ CFU) infection studies by week 14 (Figure 2). In these studies, nymphal *Dv* and *Aa* ticks were infected with *Ft* by feeding uninfected ticks on infected mice (10^7^ CFU/mouse; as described above), ticks were collected when replete, maintained for 14 weeks (including through the nymph-to-adult molt), individually placed onto naïve mice, allowed to take a blood meal, replete ticks were collected, and then mice were monitored for 21 days (or until humanely euthanized because of severe signs of tularemia). In these transmission experiments, only infected adult female ticks were placed onto naïve mice. The rationale for using adult female ticks for infection studies included: (i) adult *Dv* ticks have been reported to more efficiently transmit infections than *Dv* nymphs [18,28]; (ii) adult male ticks only partially feed on hosts before detaching and mating with adult female ticks, likely limiting the ability of adult male ticks to efficiently transmit pathogens [29,30]; (iii) in parallel experiments, we found that female ticks harbored higher *Ft* numbers than male ticks (8.1 × 10^6^ CFU/female *Dv* tick vs. 1.9 × 10^4^ CFU/male *Dv* tick; 3.2 × 10^5^ CFU/ female *Aa* tick vs. 1.8 × 10^5^ CFU/male *Aa* tick; Appendix A), although not significant. In transmission studies, we placed infected adult female ticks, along with uninfected adult male ticks, to promote attachment and efficient feeding by female ticks [29].

Infected adult female *Dv* and *Aa* ticks (week 14) were individually placed, together with a non-infected adult male tick of the same species, onto naïve mice to assess *Ft* transmission. Of the 15 adult *Dv* female ticks and 15 adult *Aa* female ticks placed onto mice, 14 *Dv* ticks (93% attachment) and 13 *Aa* ticks (66% attachment) attached and fed to repletion within 7–12 days on naïve mice. Following tick engorgement/detachment and collection (day 0), mice were bled to quantitate *Ft* numbers in individual mice, with 71% (10 of 14) of *Dv*-infested mice (mean 1.8 x 10^5^ CFU/mL) and 61% (9 of 13) of *Aa*-infested mice (mean 7.8 × 10^4^ CFU/mL) having detectable CFUs in the blood on day 0 (Figure 3). Although not significant, differences in *Ft* numbers in mouse blood on day 0 (*Dv*- vs. *Aa*-infested mice; Figure 3) appeared to correlate with differences in *Ft* numbers in infected female ticks (*Dv*- vs. *Aa*-infected ticks; Appendix A), where *Dv* ticks harbored and transmitted higher numbers of *Ft,* compared to *Aa* ticks.

After tick harvest, mice (*n* = 14 for *Dv-*infested group; *n* = 13 for *Aa*-infested group) were monitored until they developed severe signs of tularemia (hunched, ruffled fur, conjunctivitis, unable to move when gently prodded), mice were humanely euthanized, blood, skin, lungs, livers, and spleens were harvested, and bacterial burdens were enumerated from each sample. From *Dv*-infested mice, 57% (8 of 14 mice) exhibited signs of tularemia within 2 days of tick detachment, with an average time to moribund status of 1.25 days. From *Aa*-infested mice, 46% (6 of 13 mice) exhibited signs of tularemia within 2 days after tick detachment, with an average time to moribund status of 1.20 days. Interestingly, two mice (1 *Dv*-infested; 1 *Aa*-infested) reached moribund status before their respective ticks completed feeding, demonstrating the ability of infected *Dv* and *Aa* ticks to cause rapid and lethal disease in animals. Between day 0 (tick repletion) and moribind status (within 2 days of tick repletion), *Ft* replicated approx. 0.6 orders of magnitude in the blood of both *Dv*- and *Aa*-infested mice and *Ft* was detected in all tissues collected from moribund mice (Figure 3). Comparing *Ft* numbers in *Dv*- and *Aa*-infested mouse tissues, *Dv*-infested mice had higher, although not significant, bacterial burdens in blood (mean 8.4 × 10^5^ CFU/mL; Figure 3), lungs, livers, and spleens (means: 1.7 × 10^5^–5.0 × 10^5^ CFU/mg tissue; Figure 3) on days 0–2, compared to *Aa*-infested mice (blood mean 4.1 × 10^5^ CFU/mL; lungs, livers, and spleens means: 2.8 × 10^5^–3.5 × 10^5^ CFU/mg tissue; Figure 3). When comparing *Ft* numbers at the tick attachment site, *Dv*-infested mice had higher bacterial numbers, although not signficiant, in mouse skin (1.0 × 10^2^ CFU/mg), compared to *Aa*-infested mouse skin (3.9 × 10^1^ CFU/mg). Additionally, classic tularemia skin ulcerations [31] were observed at the tick attachment site in 37% (3 of 8) of *Dv*-infested mice and 33% (2 of 6) of *Aa*-infested mice (Appendix A). Although both *Dv* and *Aa* ticks transmitted rapid and lethal infections to naïve mice, higher *Ft* numbers in female *Dv* ticks in week 14 (Figure 2 and Appendix A), higher rates of *Dv*-attachment to mice, and higher bacterial burdens in blood, skin, lungs, livers, and spleens of *Dv*-infested mice (Figure 3) indicate that *Dv* ticks may pose a greater health risk for transmitting and causing more severe tularemia infections.

### 2.4. FTL1793 Exhibits Chitinase Activity

Chitin, a polymer of N-acetylglucosamine (GlcNAc), is a major component of the tick cuticle and is remodeled during the tick molting process. As such, it has been speculated that chitin may be used as an energy source by tick-borne bacterial pathogens [32,33,34]. Chitin cleavage can be performed by chitinases, and bacterial chitinases have been shown to promote bacterial persistence in marine environments and mammals [35,36,37,38]. Using a conserved domain search, we searched the *Ft* LVS genome for putative chitinases and, confirming a previous report [25], found that gene locus *FTL1793* (annotated as a hypothetical protein) contains a region encoding a putative glycosyl hydrolase (GH) 18 chitinase D-like region (Appendix A). Although some chitinase D-like proteins consist only of a catalytic domain [39,40], other chitinases are known to contain both a catalytic GH18 domain and a chitin-binding domain, linked by a fibronectin type III domain [41,42]. Different chitinase proteins are known to vary in chitin binding and chitin cleavage activity based on the presence/absence these domains [43]. *FTL1793* was found to only contain a GH18 chitinase domain.

To confirm putative chitinase activity of *FTL1793*, we cloned, expressed, and purified recombinant *FTL1793* protein. Recombinant *FTL1793* was estimated to be approx. 90% pure, based on visualization of purified protein and immunoblot analysis (Appendix A). The ability of recombinant *FTL1793* to cleave purified chitin was tested in chitin azure assays [44], where chitin has been covalently linked to remazol brilliant violet 5R (RBV) dye and chitin cleavage (dye release) can be monitored at 570 nm. Chitinase activity was measured for *FTL1793*, along with a positive control (chitinase from *Streptomyces griseus*), a negative control (*Ft* outer membrane protein FopA; [45,46]), and chitin azure assay buffer alone. An increase in absorbance was observed within 48 h in samples containing either *FTL1793* or *S. griseus* chitinase, confirming the predicted chitinase activity of *FTL1793* (Figure 4A). Given the lower relative chitinase activity of *FTL1793* during the first 24 h, compared to *S. griseus* chitinase (Figure 4A), and the likely need for sustained *Ft* chitinase activity in ticks during an overwintering, separate chitin azure assays were set up and monitored over 30 days to assess long-term *FTL1793* activity. In these long-term chitin azure assays, *FTL1793* and *S. griseus* chitinase both demonstrated sustained chitinase activity during the 30 days incubation (Figure 4B). Negative control protein FopA and buffer alone did not display chitinase activity in either the 48 h or 30 days chitin azure assays (Figure 4A,B). These results confirmed that *FTL1793* exhibits chitinase activity and suggested that *Ft* may use this chitinase to degrade chitin in ticks. Furthermore, sustained *Ft* chitinase activity (Figure 4B) may be needed for long-term persistence in ticks (e.g., overwintering event).

### 2.5. Ft FTL1793, a Putative Chitinase, Is Required for Ft Persistence in Ticks

Although previous studies speculated that chitinases may play important roles in *Ft* environmental persistence or in *Ft* tick infections [25,47], only one study has examined if chitinases are required for or contribute to *Ft* infections of ticks. However, that study co-infected *D. andersoni* ticks with a combination of three independent *Francisella novicida* chitinase mutants (*chiA, chiB, chiAB*), finding that all three mutants were present in ticks four days later, suggesting that *F. novicida* chitinases may not be required for tick infections [48]. Given the lack of studies testing *Ft* and putative *Ft* chitinases in either *Dv* or *Aa* ticks, the true role of *Ft* chitinases in tick infections remains unclear. A separate study examined an *Ft purMCD* mutant (purine auxotroph), which is avirulent in mice, in *Dv* ticks, finding that the *purMCD* mutant did not persist in ticks through the molt to the adult stage [19]. However, given the high degree of attenuation of that mutant in mice and low initial infectivity of *purMCD* in *Dv* ticks, it is unlikely that purine biosynthesis, alone, plays a major role in *Ft* persistence in ticks. To examine the role of *FTL1793* and associated chitinase activity in tick infections, an isogenic *Ft* LVS mutant was generated using homologous recombination [49], referred to hereafter as Δ*FTL1793*. To confirm that Δ*FTL1793* did not possess an inherent growth defect, the growth of WT *Ft* LVS and Δ*FTL1793* were compared in liquid growth medium, demonstrating no substantial differences in bacterial growth over 32 h (Appendix A).

Before testing the infectivity of the Δ*FTL1793* mutant in ticks, we first assessed the virulence of Δ*FTL1793* in mice. These initial mouse virulence testing studies were important because our mouse–tick–*Ft* infection model relies on *Ft* virulence, including bacteremia in mice, to reproducibly infect ticks (Figure 1 and Figure 2). Although our mouse–tick–*Ft* infection model infects mice i.v. with *Ft* two days before tick repletion, it remained possible that this two-day i.v. infection window was too narrow to assess potential virulence defects of Δ*FTL1793.* As such, we performed mouse pulmonary infections (intranasal delivery) to assess Δ*FTL1793* virulence. We, and others, have previously reported on the virulence of *Ft* LVS in mice via the intranasal route, which typically results in morbidity/mortality within 5–8 days [50,51,52]. Groups of mice were intranasally infected with either 10^4^ CFU of wild-type (WT) *Ft* LVS, 10^4^ CFU of Δ*FTL1793*, 10^7^ CFU of Δ*FTL1793*, or 10^9^ CFU of Δ*FTL1793*, and monitored for 21 days after infection (or euthanized when moribund). Confirming previous intranasal infection studies in our laboratory, all mice intranasally infected with 10^4^ CFU of WT *Ft* LVS died within seven days (Appendix A). At 10^4^ CFU, Δ*FTL1793* was partially attenuated, with 60% (three of five) of mice surviving until day 21 (Appendix A). In contrast, only 25% (one of four) of mice infected with either 10^7^ CFU or 10^9^ CFU of Δ*FTL1793* survived infection (Appendix A). Given that there was no difference in percentage survival of mice infected with either 10^7^ or 10^9^ CFU of Δ*FTL1793,* we selected the 10^7^ CFU infectious dose of Δ*FTL1793* for subsequent mouse–tick–*Ft* infection studies and to correlate these findings with our previous 10^7^ CFU infectious doses studies (Figure 2).

To assess the role of *FTL1793* in promoting *Ft* infections of ticks, *Dv* and *Aa* ticks were infected with either WT *Ft* LVS or Δ*FTL1793*, identical to what is described above for the high dose (10^7^ CFU) tick infection studies (Figure 2). In these studies, uninfected *Dv* and *Aa* ticks were placed onto naïve mice (day 5), the ticks fed for three days, mice were i.v. infected (day 2) with either 2.4 × 10^7^ CFU of WT *Ft* LVS or 3.8 × 10^7^ CFU of Δ*FTL1793,* and replete ticks and mouse blood were collected two days later (day 0). Despite partial attenuation of Δ*FTL179*3 via the intranasal route (Appendix A), Δ*FTL1793* was not attenuated via the i.v. route, because no significant differences were found between WT *Ft* LVS and Δ*FTL1793* numbers in the blood of infected mice (either *Dv*-infested or *Aa*-infested) on day 0 (Figure 5A,B). In addition, both WT *Ft* LVS and Δ*FTL1793* were found to have replicated approx. 1 order of magnitude in mouse blood over two days (Figure 5A,B).

On day 0, replete *Dv* ticks were found to be infected by nearly 1 order of magnitude more WT *Ft* LVS (mean 9.4 × 10^5^ CFU/*Dv* tick) than Δ*FTL1793* (mean 1.3 × 10^5^ CFU/*Dv* tick), although not significantly (Figure 5A). Similarly, in week 2, nearly 0.5 orders of magnitude more WT *Ft* LVS (mean 1.7 × 10^6^ CFU/*Dv* tick) was present in *Dv* ticks, compared to Δ*FTL1793* (mean 6.3 × 10^5^ CFU/*Dv* tick), although not significantly (Figure 5A). In week 4, WT *Ft* LVS was 2 orders of magnitude higher (mean 4.4 × 10^6^ CFU/*Dv* tick) than Δ*FTL1793* (mean 3.4 × 10^4^ CFU/*Dv* tick) in *Dv* ticks (*p* = 0.0002; Figure 5A). Between weeks 2 and 4, differences in WT and Δ*FTL1793* bacterial numbers in *Dv* ticks were further highlighted by a 2.5-fold replication of WT *Ft* LVS compared to a 16-fold decrease in Δ*FTL1793* in *Dv* ticks during the same time period (Figure 5A). Between weeks 4 and 6, nymphal *Dv* ticks molted to adults, with WT *Ft* LVS decreasing two-fold (mean 2.2 × 10^6^ CFU/*Dv* tick in week 6), while Δ*FTL1793* increased 650-fold (mean 2.5 × 10^7^ CFU/*Dv* tick in week 6) (Figure 5A). However, differences between WT *Ft* LVS and Δ*FTL1793* were not signficantly different in week 6.

In *Aa* ticks, approx. 0.7 orders of magnitude more WT *Ft* LVS (mean 1.7 × 10^7^ CFU/*Aa* tick) was present on day 0, compared to Δ*FTL1793* (mean 4.4 × 10^6^ CFU/*Aa* tick), although not significantly (Figure 5B). In week 2, WT *Ft* LVS (mean 2.4 × 10^6^ CFU/*Aa* tick) and Δ*FTL1793* (mean 2.2 × 10^6^ CFU/*Aa* tick) numbers were similar, with both bacterial strains having decreased in *Aa* ticks from day 0 to week 2 (Figure 5B). In week 4, *Ft* LVS numbers (mean 1.2 × 10^6^ CFU/*Aa* tick) were 2.6 orders of magnitude higher (*p* = 0.0332) than Δ*FTL1793* (mean 4.7 × 10^3^ CFU/*Aa* tick) (Figure 5B). Differences between WT *Ft* LVS and Δ*FTL1793* numbers in week 4 were highlighted by differences in replication rates between weeks 2 and 4, with WT *Ft* LVS decreasing two-fold in *Aa* ticks, while Δ*FTL1793* decreased over 460-fold (Figure 5B). In week 6, WT *Ft* LVS was >1 order of magnitude higher (mean 2.1 × 10^6^ CFU/*Aa* tick) than Δ*FTL1793* (mean 9.0 × 10^4^ CFU/*Aa* tick; *p* < 0.05), with WT *Ft* LVS having replicated 1.7-fold between weeks 4 to 6, compared with 18-fold replication of Δ*FTL1793* during the same time period (Figure 5B). Comparing Δ*FTL1793* and WT *Ft* LVS in both *Dv* and *Aa* ticks, Δ*FTL1793* was less efficient at infecting (day 0), replicating in, and persisting in nymphal ticks until week 4 (Figure 5A,B). However, after the molt to adult (week 6), Δ*FTL1793* was found in higher numbers in *Dv* ticks and replicated faster than WT *Ft* LVS in both *Dv* and *Aa* ticks. Taken together, these data indicate that *FTL1793* and its associated chitinase activity contribute to the infection and persistence of *Ft* in nymphal ticks, prior to the molt to the adult life stage (e.g., first four weeks). Increases in Δ*FTL1793* numbers after the molt indicate that additional chitinases or other genes are required for *Ft* to persist and replicate in ticks long-term.

## 3. Discussion

Although tick-borne tularemia was described as early as 1924, major questions remain about which arthropod vectors pose the greatest risk for harboring and transmitting *Ft* to humans. Data from the CDC indicate that *D. andersoni*, *Dv,* and *Aa* ticks, as well as deer flies (*Chrysops* sp.), can transmit tularemia. In addition, modeling has predicted that climate change, specifically milder winters and increased precipitation at higher latitudes, will drive major increases in tick numbers and expansion of ticks into new geographic areas in upcoming years [8,9,53,54]. Finally, the appearance of invasive tick species, including *Hl* from Asia, in the U.S. has raised concerns about the introduction of exotic tick-borne diseases into the U.S., the transmission of existing U.S. tick-borne diseases by invasive tick species, and co-transmission of two or more diseases by ticks to humans [55]. Although *Hl* ticks were shown not to vector *Borrelia burgdorferi* [56], the causative agent of Lyme disease, *Hl* ticks were shown to harbor and transmit *Rickettsia rickettsii* [57], the agent of Rocky Mountain spotted fever, leaving unanswered questions about what other tick-borne diseases can be harbored and transmitted by *Hl* ticks. Those findings are not unique to *Hl* ticks; it was previously reported that *B. burgdorferi* was unable to persist in *Dv* ticks, likely due to the presence of *Dv*-specific antimicrobial peptides that are lytic to the spirochete [58].

Despite a number of previous studies examining *Ft* infection or transmission by *Dv* ticks [15,16,18,19,28,59,60], much less information is available about *Ft* infections of *Aa* ticks [20,21]. Importantly, none of those previous studies directly compared *Ft* infection of and transmission by *Dv* and *Aa* ticks. Furthermore, given concerns about the invasion of *Hl* ticks into the U.S., no studies have examined whether *Ft* can infect *Hl* ticks. In this study, we directly compared *Ft* infections of *Dv*, *Aa*, and *Hl* nymphal ticks, at two different infectious doses, and monitored *Ft* persistence, replication, and/or clearance in these ticks over the course of 6–14 weeks. This is the first study to report that, although *Ft* initially infects Hl ticks, *Ft* is unable to persist or is cleared by *Hl* ticks, at both 10^5^ and 10^7^ CFU infectious doses. Due to the lack of *Ft* in *Hl* ticks in week 14, we were unable to perform transmission studies. However, given that a low dose (10^5^ CFU) of *Ft* was not detected in *Hl* ticks after two weeks and a high dose (10^7^ CFU) of *Ft* was not detected in *Hl* ticks after six weeks, our results indicate that *Hl* ticks are unlikely to serve as a major vector for tularemia. By directly comparing *Ft* infection and persistence in both *Dv* and *Aa* ticks, the two major U.S. tick vectors for tularemia, we found that *Aa* ticks initially were infected by approx. 1 order of magnitude more *Ft* (at both 10^5^ and 10^7^ CFU infectious doses), compared to *Dv* ticks. Higher *Ft* numbers in *Aa* ticks were not due to larger blood meal volumes by *Aa* ticks (when relative CFU/tick were adjusted for weights of engorged ticks), suggesting either that there are inherent differences in how *Ft* infects *Dv* and *Aa* ticks, or there are differences in how *Dv* and *Aa* ticks respond to *Ft* infections. Another possibility is that *Aa* ticks concentrate their blood meals more efficiently than *Dv* ticks. One previous study examined the blood volumes imbibed by various life stages of *Aa* ticks (larvae, nymph, adult), finding that *Aa* ticks remove excess ions and water to concentrate the blood meals [61]. Other studies have used isotopes to accurately measure the amount of red blood cells and plasma imbibed by other tick species [62]. However, it is not known if *Aa* ticks concentrate their blood meal more efficiently than *Dv* ticks, which could help explain differences in *Ft* numbers in engorged ticks. Separately, other studies have shown that erythrocyte invasion by *Ft* may enhance its ability to colonize ticks following a blood meal [63]. Although it is unclear how *Ft* invades erythrocytes, given that these cells do not phagocytose or endocytose, it is possible that *Aa* ticks imbibe an increased number of erythrocytes via blood meal concentration, compared to *Dv* ticks, which also may help to explain higher *Ft* numbers in engorged *Aa* ticks [63].

Alternatively, differences in *Dv* and *Aa* endosymbionts or tick midgut microbiota [64,65,66,67] may impact the ability of *Ft* to initially colonize and persist in ticks by competing with *Ft* for nutrients, providing essential nutrients to *Ft*, or modulating tick immune responses that either promote or restrict *Ft* infection. Although some previous studies have speculated that tick endosymbionts may provide nutrients to other tick-borne pathogens [64,68], nothing has been reported about how *Dv* and *Aa* endosymbionts influence the *Ft* persistence in ticks. Our studies demonstrated that *Ft* persisted in both *Dv* and *Aa* ticks for up to 14 weeks (3.5 months) following engorgement. After the tick processes its blood meal, nutrients become extremely limited [69]. For *Ft*, the lack of nutrients in ticks during an overwintering event is likely to be further confounded by missing and/or incomplete amino acid biosynthesis pathways [70], suggesting that *Ft* may need to acquire nutrients exogenously. Although it remains possible that tick endosymbionts could provide essential nutrients to *Ft* and that differences in *Dv* and *Aa* endosymbionts could explain differences in *Ft* numbers and *Ft* replication rates in either tick, detailed studies on *Dv* and *Aa* endosymbionts, in the context of *Ft* infections, are needed. Such studies could provide important information about tick–microbiome–pathogen interactions.

Despite higher initial *Ft* numbers in *Aa* ticks, we found that *Ft* replicated more robustly in *Dv* ticks, increasing 20-fold in *Dv* ticks over six weeks at the low infectious dose (compared to a 1.1-fold replication in *Aa* ticks during the same time period) and increasing 1.7-fold in *Dv* ticks over 14 weeks at the high infectious dose (compared to a 64-fold decrease in *Aa* ticks over the same time period). Although higher *Ft* replication rates in *Dv* ticks did not correlate with a significantly higher *Ft* transmission to naïve mice, *Ft* numbers in *Dv*-infested mice were generally higher in all tissues examined, compared to *Ft* numbers in *Aa*-infested mice. Interestingly, at the low infectious dose, *Ft* replicated robustly in both *Dv* and *Aa* ticks between weeks 2 and 4. Other tick researchers have speculated that the breakdown of the peritrophic matrix, a semipermeable membrane that surrounds the tick blood meal, contributes to the availability of nutrients for residing pathogens [32,33]. Because the tick peritrophic matrix is composed of chitin [71], bacteria able to degrade this N-acetyl-glucosamine polymer may be able to generate carbon as an energy source [35,72]. Given that *Dv* and *Aa* ticks molted between weeks 4 and 6, chitin may have been available preceding the molt (between weeks 2 and 4), which may explain the 11-fold *Ft* replication in *Dv* ticks and 5-fold *Ft* replication in *Aa* ticks between weeks 2 and 4. In contrast, *Ft* replication between weeks 2 and 4 in the high infectious dose study was modest in both *Dv* and *Aa* ticks. One possibility is that the amount of free chitin in ticks before the molt is limited and higher *Ft* numbers consumed all available chitin. Indeed, in these higher dose infections, bacterial burdens did not exceed 10^7^ CFU/tick, suggesting that there is a limit to *Ft* numbers in ticks.

Other pathogenic bacteria, including *Vibrio cholerae*, have been shown to utilize chitinases to promote environmental persistence and form biofilms on chitin in the marine environment [36,73]. Similarly, *F. novicida* was reported to form biofilms on chitin surfaces and use chitin as a sole energy source. In that study, two *F. novicida* chitinase mutants, Δ*chiA* (*F. novicida* gene locus *FTN_0627*) and Δ*chiB* (*F. novicida* gene locus *FTN_1744*), were unable to colonize or form biofilms on chitin [47]. However, given the many differences between *Ft* and *F. novicida* [74,75], and previously-noted differences among *Ft* Type A, *Ft* Type B, and *F. novicida* putative chitinases [25], the application of those findings to *Ft* strains remains unknown. Considering those studies, and that chitin is a major component of ticks (including the peritrophic matrix), we hypothesized that *Ft* chitinases may play important roles in infection and persistence in ticks. Of the four putative *Ft* chitinases, ChiA (gene loci *FTT_0715* in Type A strain SchuS4, *FTL_1521* in LVS, and *FTN_0627* in *F. novicida*), ChiB (gene loci *FTT_1768* in SchuS4, *FTL_0093* in LVS, and *FTN_1744* in *F. novicida*), and ChiC (gene loci *FTT_1592c* and *FTT_1593c* in SchuS4, *FTL_1635* in LVS, and *FTN_0627* in *F. novicida*) were reported to vary substantially among *Francisella* sp. in amino acid identity, predicted open reading frame lengths, and predicted domains within each gene [25]. Here, we focused on ChiD (gene loci *FTT_0066* in SchuS4, *FTL_1793* in LVS, and *FTN_1644* in *F. novicida*) given that *chiD* sequences were highly conserved (98.6% identity) among *Ft* Type A, *Ft* Type B, and *F. novicida*. Our Δ*FTL1793* mutant, lacking a GH18 chitinase, was detected at significantly lower numbers (2 orders of magnitude less in *Dv* ticks; 2.5 orders of magnitude less in *Aa* ticks) in ticks in week 4 (immediately before the tick molt). In addition, our in vitro chitinase activity assays confirmed that recombinant *FTL1793* continuously cleaved chitin (throughout 30-day incubation). Taken together, these results suggest that *FTL1793* degrades tick chitin, likely liberated immediately prior to tick molting, to promote *Ft* replication in both *Dv* and *Aa* ticks. Despite significantly lower Δ*FTL1793* numbers in week 4, Δ*FTL1793* numbers subsequently increased in both ticks between weeks 4 and 6, suggesting that other proteins, including other chitinases, may provide nutrients to *Ft* to promote bacterial replication. As noted above, a previous study noted that *Ft* contains at least four chitinases (ChiA, ChiB, ChiC, ChiD) and speculated that different chitinases may function at different stages throughout *Ft* tick infection [25,47]. Interestingly, in that previous study, none of the ChiD orthologs (including gene loci *FTT_0066* in SchuS4, *FTL_1793* in LVS, and *FTN_1644* in *F. novicida)* were found to possess chitinase activity. However, those authors tested chitinase activity using different chitin substrates, including chitin analogs, and incubation times and conditions also varied from our studies. Regardless, given our findings that *FTL1793* exhibits chitinase activity and *FTL1793* contributes to persistence and replication of *Ft* in both *Dv* and *Aa* ticks, future studies are needed to fully elucidate the function of all *Ft* chitin-related genes in ticks.

Finally, it remains possible that *FTL1793*, and other *Ft* chitinases, may perform functions in addition to degrading arthropod/tick chitin. In *Pseudomonas aeruginosa*, a pulmonary pathogen, endochitinase activity correlates with antifungal activity, which may be important for microbial competition in the lung [76]. In *Legionalla pneumophila*, another pulmonary pathogen, the ChiA chitinase is required to cleave lung mucin and promote bacterial penetration through the alveolar mucosa [38]. Other bacterial chitinases have been shown to play similar roles in bacterial virulence, including cleavage of mammalian glycolipids, glycoproteins, and extracellular matrix components [77]. Here, our initial virulence testing of the Δ*FTL1793* mutant in a mouse pulmonary infection model revealed that Δ*FTL1793* was partially attenuated. However, Δ*FTL1793* bacteremia numbers (i.e., intravenous infection) were equivalent to WT *Ft* LVS after two days of infection, indicating that *FTL1793* may be required for *Ft* virulence in lungs but not in the bloodstream. Given these findings, future studies should examine the potential multi-functional role(s) of *FTL1793*, and other *Ft* chitinases, in both ticks and mammalian hosts, and consider different infection routes for these studies.

## 4. Materials and Methods

### 4.1. Bacterial Stains and Culture Conditions

*Francisella tularensis* Type B strain LVS was obtained from BEI Resources and cultured as previously described [50,51]. Routine *F. tularensis* cultures were grown overnight on supplemented Mueller-Hinton agar (sMHA) at 37 °C with 5% CO_2_. sMHA plates were prepared with the following: Mueller-Hinton broth powder (Becton Dickinson) was mixed with 1.6% (wt/vol) Bacto Agar (Becton Dickinson), autoclaved, and further supplemented with 2.5% (vol/vol) bovine calf serum (Hyclone), 2% (vol/vol) IsoVitaleX (Becton Dickinson), 0.1% (wt/vol) glucose, and 0.025% (wt/vol) iron pyrophosphate. For mouse infections, *F. tularensis* strains were first grown for 24 h on sMHA then transferred to Brain heart infusion agar (BHI; Becton Dickinson) and grown overnight at 37 °C with 5% CO_2_.

### 4.2. Mouse–Tick–F. tularensis Infection and Transmission

*Dermacentor variabilis*, *Amblyomma americanum,* and *Haemaphysalis longicornis* nymphal and adult ticks were obtained from either the Centers for Disease Control and Prevention through the Biodefense and Emerging Infectious Diseases (BEI) Resources Repository or from the Tick Rearing Facility, National Tick Research and Education Resource, Oklahoma State University, Stillwater, OK, U.S.A. Ticks were housed in 3-dram plastic vials in a glass desiccator, with 12 h light–dark cycles, at ambient temperature over saturated potassium nitrate (KNO_3_), which generated a humidified atmosphere of >90% at 20 °C. All studies with mice and ticks were approved by the University of Toledo Institutional Animal Care and Use Committee (IACUC; protocol 108672 approved May 2019 and valid until May 2022) and Institutional Biosafety Committee (IBC; protocol 108665 approved March 2016 and valid until March 2021). C3H/HeN mice, female, 8–10 weeks old, were purchased from Charles River Laboratories. One day prior to tick placement (day 6), mice were anesthetized with an i.p. injection of ketamine-xylazine, an area approximately 2.5 cm in diameter between the shoulder blades was shaved with surgical clippers, and plastic chambers (top portion of 15 mL conical tubes) were adhered to shaved skin using Kamar adhesive. Mice were individually housed to prevent chamber removal by cage mates and mice were maintained in disposable, plastic cages with sealable lids and high-efficiency particulate air (HEPA) filters (Innovive). The next day (day 5), mice were anesthetized, 12 nymphal *D. variabilis*, *A. americanum* or *H. longicornis* ticks were placed in each chamber, chambers were closed with a fine-mesh polyester fabric, and fabric was secured to each chamber using a rubber band. Double-sided tape was adhered to the inside upper rim of each cage bottom, cage lids were secured onto each cage, and cages were placed onto tack mats to prevent the loss of any escaped ticks. All tick studies were performed in designated BSL2/A-BSL2 rooms with double-sided tape placed around the entry/exit door and tack mats placed in front of entry/exit doors to prevent the loss of any escaped ticks. For mouse infections, overnight LVS growth was scraped from BHI agar plates, suspended in sterile PBS, and diluted to either 10^5^ or 10^7^ CFU/100 µL, based on previous OD_600_ measurements and bacterial enumeration studies. Three days after tick placement (day 2), mice were anesthetized by an i.p. infection of ketamine-xylazine, and intravenously infected (retro-orbital) with *F. tularensis* LVS. Bacterial inocula were serially diluted and plated in quadruplets onto sMHA to confirm CFUs. Approximately 48 h after infection (day 0), mice were anesthetized by an i.p. injection of ketamine-xylazine, replete ticks were collected, and blood was harvested from infected mice by cardiac puncture for serial dilution in PBS and plated onto sMHA. Individual mice and groups of replete ticks from each mouse were sequentially numbered so that bacterial numbers could be correlated between mouse blood and associated ticks. Prior to tick homogenization, ticks were individually weighed, and weights recorded for subsequent analysis. Ticks were processed on day 0 (replete/engorgement), week 2, week 4, week 6, and week 14 after harvesting from mice. Prior to homogenization, ticks were surface-sterilized by being placed into 30% H_2_O_2_ for five seconds, 70% ethanol for five seconds, rinsed with molecular biology grade water (Corning) for 5 s, then homogenized in RNase-free disposable pellet pestle tubes (Fisher) containing 200 µL of sterile PBS. Tick homogenates were serially diluted in PBS and plated onto sMHA containing 100 mg/L cycloheximide, 80,000 U/L polymyxin B, and 2.5 mg/L amphotericin B. Following 72 h of incubation, colonies were counted, and CFU/mL (mouse blood), CFU/tick, or CFU/mg tick were calculated. Transmission studies (from infected ticks to naïve mice) were performed essentially as described above, with the following modifications: LVS-infected adult female ticks (14 weeks after their nymphal blood meal) were individually placed into chambers and allowed to feed to repletion on naïve mice. A single non-infected adult male tick of the same species was added to each chamber to promote female tick attachment and efficient feeding. Replete adult ticks were collected 7–12 days after tick attachment. To quantitate bacterial burdens in mouse blood during transmission studies, mouse blood was collected after ticks completed their blood meal (7–12 days after tick placement) via retro-orbital bleeding. Following tick detachment, mice were monitored at least three times daily for signs of tularemia and were humanely euthanized when moribund (within 2 days of tick detachment). For euthanasia, mice were anesthetized by an i.p. infection of ketamine-xylazine, blood was collected by cardiac puncture, mice were cervically dislocated, skin from the tick attachment site was harvested using an 8 mm biopsy punch (Accuderm, Fort Lauderdale, FL, USA), and lungs, livers, and spleens were aseptically harvested and transferred to sterile Whirlpack (Madison, WI, USA) bags. Skin, lungs, livers, and spleens were homogenized, 25 µL of PBS/mg of tissue was added to each tissue, serially diluted, and dilutions were plated onto sMHA. After 72 h of incubation at 37 °C, bacterial numbers were enumerated from all samples. Fold changes in CFU/tick between time points were calculated by the formula: final value/initial value (Y/X).

### 4.3. Bioinformatic Predictions

A putative chitinase domain in gene locus *FTL1793* was identified using the National Center for Biotechnology Information (NCBI) Conserved Domain Search: https://www.ncbi.nlm.nih.gov/Structure/cdd/wrpsb.cgi).

### 4.4. Generation of F. tularensis Gene Deletion Mutant

Isogenic deletion mutants in *F. tularensis* were generated by homologous recombination as previously described [49]. *F. tularensis* LVS genomic DNA was extracted using phenol-chloroform (Fisher Bioreagents, Chicago, IL, USA). Approx. 700 bp regions immediately upstream and downstream from gene locus *FTL1793* were PCR-amplified from *F. tularensis* Type B strain LVS genomic DNA using the following primers, respectively: *FTL1793*_A (5′-GGTAAGGGGCCCGCTTTTAACTGACTTGAAGCC-3′) and *FTL1793*_B (5′-AACTTCCGCCGGCGTAGTATCGCCAGATTCATTCATTTCC-3′); *FTL1793*_C (5′-AACTTCCGCCGGCGTAGTAAACCCTTGTTGTTGAACCTG-3′) and *FTL1793*_D (5′-GGTAAGGGGCCCAGCTGATATGGTGAGTCTGC-3′). Separately, a flippase (FLP) recombination target (FRT)-flanked Pfn-kanamycin resistance cassette (*kan*) was PCR-amplified from pLG66a, as previously described [49]. Splicing overlap extension (SOE) PCR was performed to join the upstream and downstream flanking regions with the FRT-Pfn-*kan*-FRT amplicon, which replaced *FTL1793*. The resulting upstream-FRT-Pfn-*kan*-FRT-downstream insert was digested with ApaI (New England Biolabs, Ipswich, MA, USA) and ligated into similarly digested pTP163 (suicide plasmid) using T4 DNA ligase (New England Biolabs, Ipswich, MA, USA). The resulting gene deletion construct was transformed into NEB 10-β *Escherichia coli* cells (New England Biolabs, Ipswich, MA, USA) and DNA sequencing was performed to verify the integrity of the insert. Positive constructs were transformed into *E. coli* S17-1 cells and conjugation was performed with *F. tularensis* LVS on chocolate agar (Mueller-Hinton medium supplemented with 1% (wt/vol) tryptone, 0.5% (wt/vol) NaCl, 1.6% (wt/vol) agar, 1% (wt/vol) bovine hemoglobin powder (Neogen, Lansing, MI, USA), 0.1% (wt/vol) glucose, and 2% (vol/vol) IsoVitaleX (Becton Dickinson, Franklin Lakes, NJ, USA) overnight at 30 °C with 5% CO_2_. The next day, conjugants were transferred onto chocolate agar supplemented with 200 mg/L hygromycin and 100 mg/L polymyxin B, incubated for 3 to 4 days, individual colonies were transferred to sMHA supplemented with 10 mg/L kanamycin (sMHA-kan10), and kanamycin-resistant colonies were transferred to sMHA supplemented with 10 mg/L kanamycin and 8% (wt/vol) sucrose. Final replica plating of individual clones was performed by transferring colonies onto sMHA containing either 200 mg/L hygromycin (sMHA-hyg200) or sMHA-kan10. Hyg-sensitive and kan-resistant colonies, indicating loss of the suicide plasmid and replacement of *FTL1793* with *kan*, were sequence verified. The resulting gene deletion strain was designated Δ*FTL1793.*

### 4.5. Intranasal Mouse Infections

Both *F. tularensis* LVS and Δ*FTL1793* were grown on sMHA overnight, then transferred to BHI for 20–24 h. Bacteria were scraped, resuspended in sterile PBS, and diluted to the desired concentration (10^4^ to 10^9^ CFU/20 μL) based on previous OD_600_ measurements and bacterial enumeration studies. To confirm infectious doses, bacterial inocula were serially diluted and plated in quadruplets on sMHA. Groups of female C3H/HeN mice (6–8 weeks old; Charles River Laboratories, Wilmington, MA, USA) were anesthetized by an i.p. infection of ketamine-xylazine and intranasally (i.n.) infected with 20 μL of the indicated bacterial resuspension. Mice were monitored daily for signs of disease and the individual health status of all mice was recorded using 5-point health status scale (‘1′ indicated healthy mice, and ‘5′ indicated mice found dead). Moribund mice (health status ‘4′) were humanely euthanized to minimize suffering.

### 4.6. Expression and Purification of Recombinant FTL1793 Protein

Gene locus *FTL1793,* without the amino-terminal signal sequence (amino acid residues 1–32), was PCR-amplified using LVS genomic DNA, AccuPrime DNA Polymerase (ThermoFisher, Chicago, IL, USA), and primers 5′*FTL1793*_SalI (5′-GCGCGTCGACAGGAGGAAACGGATGAAGTCACTACTACCGAATAGAACAATTG-3′) and 3′*FTL1793*_BamHI (5′-GCGCGGATCCTTAGTGGTGATGGTGATGATGTTTACTATCTATTTTTGTCCAAGCATCTG-3′). Primer 3′*FTL1793*_BamHI encoded a 6 × histidine fusion tag at the 3′ end, which was added to the C-terminal end of recombinant *FTL1793* to aid in affinity purification. The resulting amplicon was double-digested with SalI and BamHI, ligated into similarly digested pBad18 using T4 DNA ligase, and transformed into NEB 10-β *E. coli* cells. Following overnight selection on Luria-Bertani (LB) agar plates containing 100 µg/mL ampicillin, individual colonies were selected, plasmids purified using Qiagen (Germantown, MD, USA) QIAprep Spin Miniprep kits, and diagnostic PCR was performed to confirm insert presence and correct size. DNA sequencing was performed for positive clones to confirm the integrity of the insert and verified expression plasmids were transformed into Rosetta DE3 *E. coli* cells (Millipore, Burlington, MA, USA) for recombinant protein expression.

To express recombinant *FTL1793* protein, bacteria were grown to an OD_600_ of 0.5 in LB medium supplemented with 100 µg/mL ampicillin, and protein expression was induced by the addition of arabinose to a final concentration of 0.4%. After 4 h of protein induction, bacteria were pelleted by centrifugation at 8000× *g* for 30 min at 4 °C, supernatant was removed, and pellets were stored at −80 °C until processing. Bacterial pellets were thawed, suspended in soluble extraction buffer (10 mM Tris, 500 mM NaCl, 10 mM imidazole, 1 mM PMSF, pH 8.0), sonicated on ice for 5 min with 30 sec intervals at 50% power, insoluble material was removed by centrifugation at 8000× *g* at 15 °C for 30 min, and supernatants were collected. Supernatants were applied to pre-equilibrated nickel-nitrilotriacetic acid (Ni-NTA) agarose (Qiagen, Germantown, MD, USA) columns, columns were washed with >10-fold excess volume of soluble extraction buffer (4 °C), and purified protein was eluted in 1 mL fractions with elution buffer (10 mM Tris, 500 mM NaCl, 200 mM imidazole, 1 mM PMSF, pH 8.0, 4 °C). Individual elution fractions were assessed by sodium dodecyl sulphate polyacrylamide gel electrophoresis (SDS-PAGE) and Coomassie staining for recombinant protein concentration and purity, with four elution fractions selected for concentration and buffer exchange (10 mM Tris, 500 mM NaCl, pH 8.0) using Amicon Ultra-4 centrifugal filter units with a 50-kDa cutoff (Millipore, Burlington, MA, USA). The final recombinant protein concentration was determined using the detergent compatible (DC) protein assay (BioRad, Hercules, CA, USA) and purity was subsequently assessed by SDS-PAGE Coomassie staining and immunoblot analysis.

### 4.7. Immunoblotting

Immunoblot analysis was performed as previously described [45]. Briefly, recombinant *FTL1793* was diluted in SDS-PAGE loading buffer, boiled for 10 min, and loaded onto 12.5% SDS-PAGE gels, together with a molecular mass standard (Precision Plus protein all blue prestained protein; BioRad Laboratories, Hercules, CA, USA). Proteins were separated, transferred to nitrocellulose, blots were incubated overnight in blot block (0.1% [vol/vol] Tween 20 and 2% [wt/vol] bovine serum albumin in PBS) at 4 °C, and immunoblotting was performed using a 1:10,000 dilution (in blot block) of Penta-His HRP conjugate antibody (Qiagen, Germantown, MD, USA). Immunoblots were developed with SuperSignal West Pico chemiluminescent detection reagent (Thermo Fisher, Rockford, IL, USA).

### 4.8. Enzymatic Assays for FTL1793 Activity

Independent chitin azure assays were prepared by adding 10 mg of chitin azure (Sigma, St. Louis, MO, USA) to 750 μL of 200 mM sodium phosphate buffer, pH 7.0, followed by one of the following: 94 μg of recombinant *FTL1793*; 94 µg of recombinant FopA (negative control *Ft* outer membrane protein; [45]); 200 U of purified chitinase from *Streptomyces griseus* (Sigma, St. Louis, MO, USA); or buffer alone (buffer control). Recombinant FopA was prepared as previously described [45]. Enzyme assays were prepared in triplicates and incubated end-over-end at 37 °C. Enzyme activity was assessed every 24 h for 30 d, with samples being centrifuged at 7000× *g* for 10 min, and the supernatant absorbance at 570 nm being measured. After absorbance measurements were recorded, samples were resuspended and returned to the incubated carousel for further analysis.

### 4.9. Statistics

GraphPad (San Diego, CA, USA) Prism 8 was used to compile data into graphs and was used for the following statistical analyses: differences in bacterial burdens in ticks and mouse blood (unpaired *t*-test; one-way ANOVA); differences in tick weight (unpaired *t*-test; one-way ANOVA); and percent survival of intranasal *F. tularensis* mouse infection (log-rank Mantel-Cox test).

## Figures and Tables

**Figure 1 pathogens-09-01037-f001:**
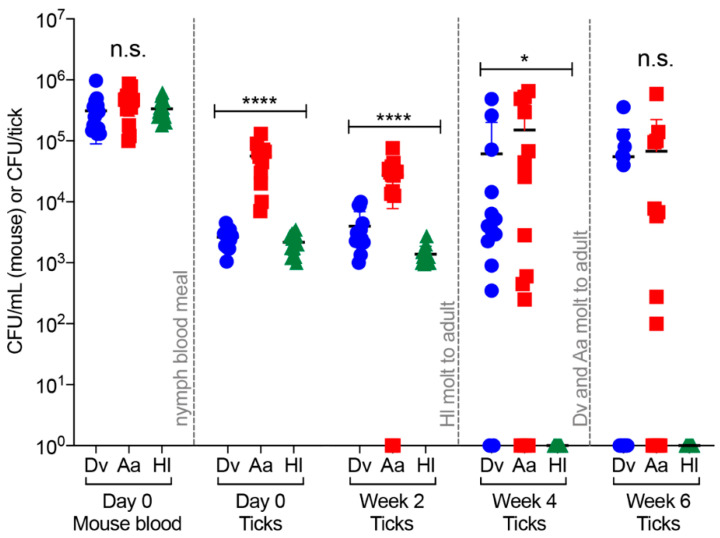
Infection, persistence, and replication of *Francisella tularensis* (*Ft*) in ticks after feeding on mice infected with 10^5^ CFU. Nymphal *Dermacentor variabilis* (*Dv*)*, Amblyomma americanum* (*Aa*), and *Haemaphysalis longicornis* (*Hl*) ticks were placed onto non-infected C3H/HeN mice (day 5), the ticks fed for 3 days, mice were i.v. infected with 10^5^ CFU of *Ft* LVS (day 2), and replete ticks were harvested 2–3 days later (day 0; 5–6 day total blood meal). Following tick harvest (day 0 ticks), blood was collected and plated from infected mice (day 0 mouse blood; *n* = 6–9/tick species/experiment) to enumerate bacterial numbers (CFU/mL blood). At the indicated time points, individual ticks (*n* = 6–9/tick species/experiment) were homogenized and plated to enumerate bacterial numbers (CFU/tick). Two independent experiments were performed to confirm reproducibility, with combined data shown. One-way ANOVA was used to compare groups of mice or ticks at each time point: n.s. indicates not significant; * indicates *p* < 0.05; **** indicates *p* < 0.0001.

**Figure 2 pathogens-09-01037-f002:**
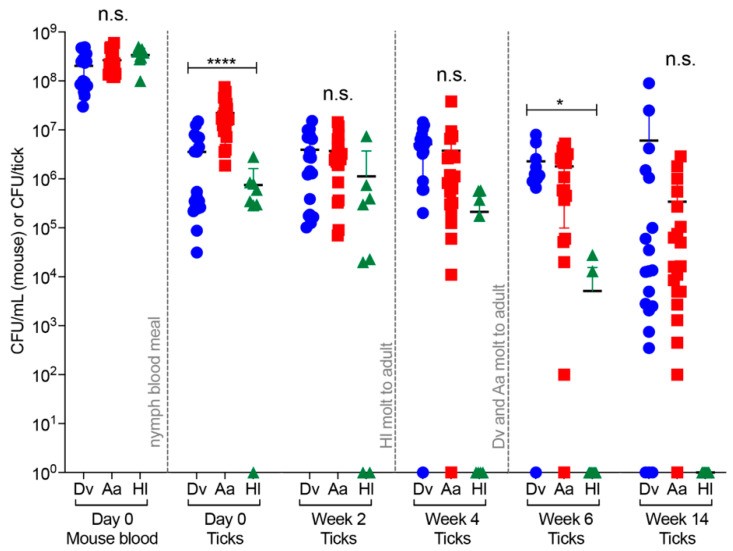
Infection, persistence, and replication of *F. tularensis* in ticks after feeding on mice infected with 10^7^ CFU. Nymphal *Dv, Aa*, and *Hl* ticks were placed onto non-infected C3H/HeN mice (day 5), ticks fed for 3 days, mice were i.v. infected with 10^7^ CFU of *Ft* live vaccine strain (LVS) (day 2), and replete ticks were harvested 2–3 days later (5–6 day total blood meal). Following tick harvest (day 0 ticks), blood was collected and plated from infected mice (day 0 mouse blood; *n* = 5–12/tick species/experiment) to enumerate bacterial numbers (CFU/mL blood). At the indicated time points, individual ticks (*n* = 7–13/tick species/experiment) were homogenized and plated to enumerate bacterial numbers (CFU/tick). Two independent experiments were performed to confirm reproducibility, with combined data shown. One-way ANOVA was used to compare groups of mice or ticks at each time point: n.s. indicates not significant; * indicates *p* < 0.05; **** indicates *p* < 0.0001.

**Figure 3 pathogens-09-01037-f003:**
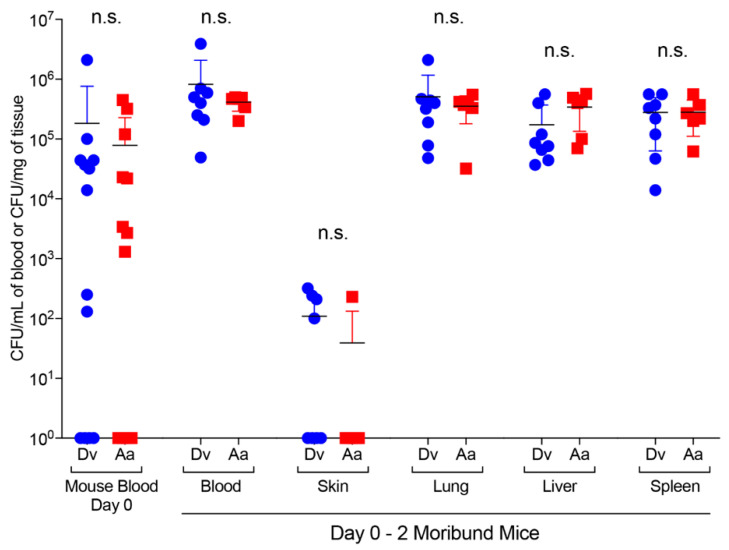
Transmission of *Ft* from infected *Dv* and *Aa* ticks to naïve mice. *Ft*-infected *Dv* and *Aa* adult ticks were prepared as described for the 10^7^ CFU infection studies (Figure 2). After 14 weeks, *Ft*-infected *Dv* and *Aa* adult ticks, together with an uninfected adult male tick of the same species, were individually placed onto naïve C3H/HeN mice and ticks were allowed to take a blood meal until replete (7–12 days). Upon repletion (day 0), ticks were harvested, and a small volume of mouse blood was collected from each mouse via retro-orbital bleeding to enumerate bacterial burdens in mice. Mice were monitored daily for signs of disease and humanely euthanized when moribund (within 2 days of tick repletion). Upon euthanasia, mouse blood and the indicated tissues (skin at attachment site, lungs, livers, and spleens) were collected and plated to enumerate bacterial numbers. Student’s *t*-test was used to compare bacterial numbers in blood samples or indicated samples: n.s. indicates not significant.

**Figure 4 pathogens-09-01037-f004:**
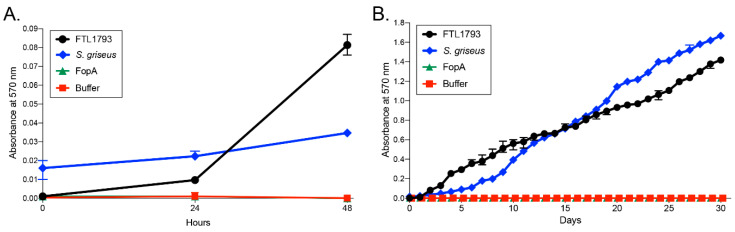
*FTL1793* exhibits chitinase activity. Recombinant *FTL1793* protein was expressed and purified from *Escherichia coli*. Chitin azure assays were used to measure chitinase activity. In these assays, 94 μg of each of the following proteins were added per reaction: recombinant *FTL1793*, chitinase from *Streptomyces griseus* (positive control), or recombinant *Ft* FopA protein (negative control). Chitin azure in assay buffer alone (buffer) was also tested. Reactions were incubated on an end-over-end carousel for either: (**A**) 48 h at 37 °C; or (**B**) 30 d at 37 °C. Absorbance at 570 nm was measured to determine chitin cleavage. Samples were prepared in triplicate and measurements recorded every 24 h.

**Figure 5 pathogens-09-01037-f005:**
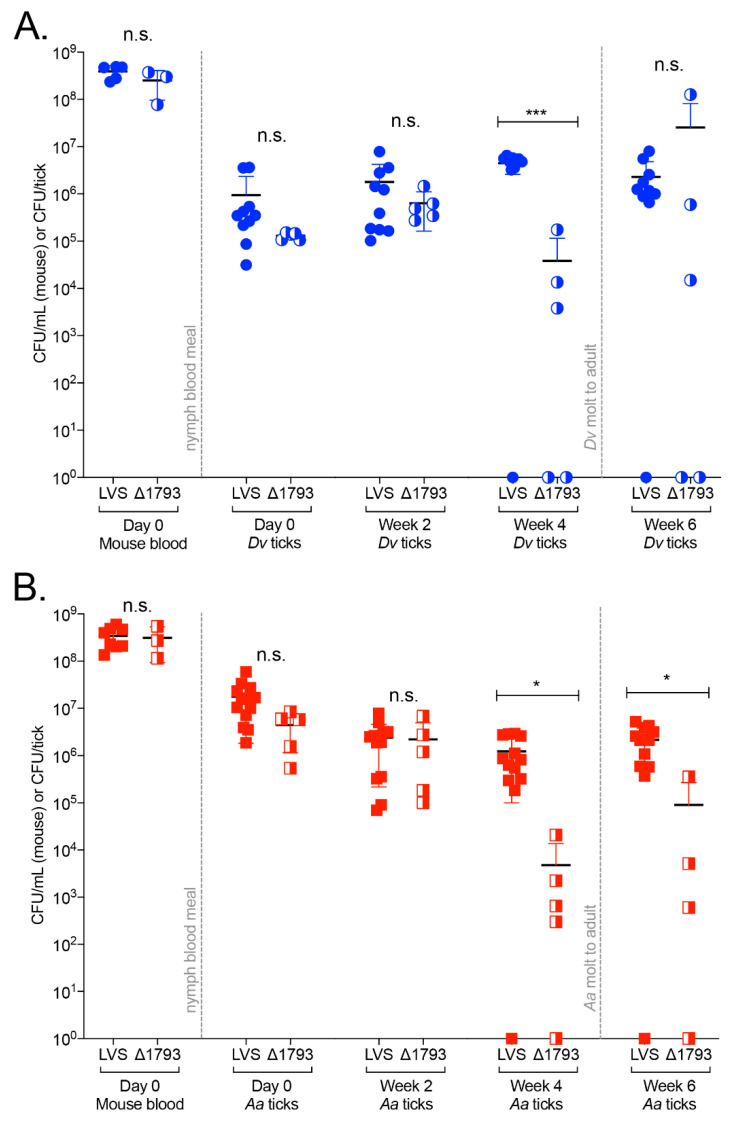
Comparison of *Ft* LVS and Δ*FTL1793* in *Dv* and *Aa* ticks. Nymphal *Dv* (panel (**A**); top; blue circles) or *Aa* (panel (**B**); bottom; red squares) ticks were placed onto non-infected C3H/HeN mice (day 5), the ticks fed for 3 days, mice were i.v. infected with 10^7^ CFU of *Ft* LVS or the Δ*FTL1793* mutant (day 2), and replete ticks were harvested approx. 2 days later (day 0; 5-day total blood meal). Following tick harvest (day 0 ticks), blood was collected and plated from infected mice (day 0 mouse blood; *n* = 3–8/group) to enumerate bacterial numbers (CFU/mL blood). At the indicated time points, individual ticks (*n* = 4–13/tick species) were homogenized and plated to enumerate bacterial numbers (CFU/tick). Student’s *t* test was used to calculate differences between LVS and Δ*FTL1793* at each time point: n.s. indicates not significant; * indicates *p* < 0.05; *** indicates *p* < 0.001.

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
