# Peer review of "A Francisella tularensis Chitinase Contributes to Bacterial Persistence and Replication in Two Major U.S. Tick Vectors"

_pathogens, 2020, doi:10.3390/pathogens9121037_

Round 1
Reviewer 1 Report
The manuscript entitled "A Francisella tularensis Chitinase Contributes to Bacterial Persistence and Replication in Two Major U.S. Tick Vectors" seemed to me very interesting, and I have almost no comments to it.
I found only one aspect that confuses me a little. Strain ΔFTL1793 maintained its growth rate on nutrient media, but it decreased its virulence in mice.
It seems that the decrease in the persistence of ΔFTL1793 in ticks is a consequence of the decrese in its ability to survive inside the organism of any host, whether it be an arthropod or a warm-blooded animal. May be, it would be worth to add a couple of phrases about the hypothesized mechanism of such a decrease in virulence in mice caused by deletion of the chitinase gene. Also, it would be interesting to add a comparison of the growth rate of the WT-strain and the mutant strain in mammalian cell cultures (if it is possible) to confirm that the growth rate does not change in hosts where there is no need to hydrolyze chitin
Author Response
Reviewer 1 comment 1: “Strain ΔFTL1793 maintained its growth rate on nutrient media, but it decreased its virulence in mice…May be, it would be worth to add a couple of phrases about the hypothesized mechanism of such a decrease in virulence in mice caused by deletion of the chitinase gene.”
Author response: We appreciate the reviewer’s comment and have added a new paragraph at the end of the discussion to address this point and propose potential mechanisms. Please see lines 631-644 of the revised manuscript.
Reviewer 1 comment 2: “Also, it would be interesting to add a comparison of the growth rate of the WT-strain and the mutant strain in mammalian cell cultures (if it is possible) to confirm that the growth rate does not change in hosts where there is no need to hydrolyze chitin.”
Author response: Although not included in the manuscript, the infectivity and intracellular growth of WT Ft LVS and our FTL1793 mutant was compared in bone marrow-derived mouse macrophages, finding no significant difference in bacterial numbers in macrophages at multiple time points. Given this negative data, we did not include it in the manuscript. We do respectfully remind the reviewer that in Figure 5, day 0 mouse blood, our FTL1793 mutant was found to intravenously infect mice and replicate in mouse blood over two days similarly as WT Ft LVS (no significant difference between groups). As such, the only mammalian infection/growth defect we observed was following a lung infection. Given this difference, as noted above, we have added a new paragraph at the end of the discussion to address this point and propose potential mechanisms.
Reviewer 2 Report
The manuscript entitled „A Francisella tularensis Chitinase Contributes to Bacterial Persistence and Replication in Two Major U.S. Tick Vectors“ by Tully and Huntley is very nicely written and readable. In their study, the authors try to answer the questions of which tick vector represents the greatest risk of tularemia transmission and which Ft protein contributes to Francisella persistence in ticks. The authors compare two U.S. tick species and one invasive species from Asia. Furthermore, by searching the Ft LVS genome, the authors found the FTL_1793 gene locus encoding a protein with chitinase activity that could be used by Ft to obtain energy by degrading chitin prior tick molting.
My question to the authors is : Is there any other gene locus encoding a protein with chitinase activity in the LVS genome? (Definitely yes.) Why did you choose FTL_1793 for further studies?
Minor comments:
- Lines 20 - 21:"We identified a putative Ft chitinase, FTL1793......" I guess this corresponds to protein which should not be written in italics according to gene/protein nomenclature guidelines.
- Line 483: There should probably be written Figures 5A and 5B, not 4B.
- Line 571:The authors mention F. novicida mutants and use the locus tags for LVS (for F. novicida the locus tag is FTN). FTL1521 and FTL0093 are LVS homologs to F. novicida FTN_0627 and FTN_1744.
- Lines 659, 674, 689: FTL1793 should be written in italics (corresponds to gene locus).
Author Response
Reviewer 2 comment 1: “Is there any other gene locus encoding a protein with chitinase activity in the LVS genome? (Definitely yes.) Why did you choose FTL_1793 for further studies?”
Author response: The abstract (lines 20-21), introduction (lines 113-118), and discussion (lines 594-613) have been modified to clarify that four putative Ft chitinases exist, to include chitinase gene loci numbers for Ft Type A, Ft Type B, and F. novicida, and provide a rationale for studying FTL_1793.
Reviewer 2, minor comment 1: Lines 20 - 21:"We identified a putative Ft chitinase, FTL1793......" I guess this corresponds to protein which should not be written in italics according to gene/protein nomenclature guidelines.
Author response: Corrected as suggested. Please note that this sentence has been revised based on the reviewer’s previous comments about explaining rationale for chitinase section.
Reviewer 2, minor comment 2: Line 483: There should probably be written Figures 5A and 5B, not 4B.
Author response: Corrected as suggested. This is line 505 of the revised manuscript.
Reviewer 2, minor comment 3: Line 571: The authors mention F. novicida mutants and use the locus tags for LVS (for F. novicida the locus tag is FTN). FTL1521 and FTL0093 are LVS homologs to F. novicida FTN_0627 and FTN_1744.
Author response: Corrected as suggested. Please note that we have revised this section of the discussion (lines 594-613) based on suggestions from both reviewers. Gene loci for Ft Type A, Ft Type B, and F. novicida are now provided for all putative chitinases.
Reviewer 2, minor comment 4: Lines 659, 674, 689: FTL1793 should be written in italics (corresponds to gene locus).
Author response: Corrected as suggested. These italicized corrections are on lines 699, 714, and 729 of the revised manuscript.
Round 2
Reviewer 1 Report
I am completely satisfied with the current version of manuscript